# Proteomics, Personalized Medicine and Cancer

**DOI:** 10.3390/cancers13112512

**Published:** 2021-05-21

**Authors:** Miao Su, Zhe Zhang, Li Zhou, Chao Han, Canhua Huang, Edouard C. Nice

**Affiliations:** 1State Key Laboratory of Biotherapy and Cancer Center, West China Hospital, and West China School of Basic Medical Sciences & Forensic Medicine, Sichuan University, and Collaborative Innovation Center for Biotherapy, Chengdu 610041, China; 2019224065144@stu.scu.edu.cn (M.S.); scuzz@stu.scu.edu.cn (Z.Z.); 2015224060079@stu.scu.edu.cn (L.Z.); 2017141494205@stu.scu.edu.cn (C.H.); 2Department of Biochemistry and Molecular Biology, Monash University, Clayton, VIC 3800, Australia

**Keywords:** personalized/precision medicine, proteomics, cancer, biomarkers, microbiome, HUPO

## Abstract

**Simple Summary:**

Cancer, a major public health issue worldwide, is the second most common cause of death. Initiatives such as the Human Genome Project (HGP) and Human Proteome Project (HPP) have greatly advanced the understanding of human health and disease, including cancer, and are supporting the current trend towards personalized/precision medicine. In this review, we will overview recent technological achievements, the key hallmarks of cancer, and unmet clinical needs. We will specifically detail the importance of cancer biomarkers in diagnosis and treatment, the role of the microbiome in health and disease, the potential of emerging omics technologies and the goals of personalized/precision medicine. Finally, we will discuss future perspectives, both from the standpoint of perceived advances in treatment, but also from the hurdles that have to be overcome.

**Abstract:**

As of 2020 the human genome and proteome are both at >90% completion based on high stringency analyses. This has been largely achieved by major technological advances over the last 20 years and has enlarged our understanding of human health and disease, including cancer, and is supporting the current trend towards personalized/precision medicine. This is due to improved screening, novel therapeutic approaches and an increased understanding of underlying cancer biology. However, cancer is a complex, heterogeneous disease modulated by genetic, molecular, cellular, tissue, population, environmental and socioeconomic factors, which evolve with time. In spite of recent advances in treatment that have resulted in improved patient outcomes, prognosis is still poor for many patients with certain cancers (e.g., mesothelioma, pancreatic and brain cancer) with a high death rate associated with late diagnosis. In this review we overview key hallmarks of cancer (e.g., autophagy, the role of redox signaling), current unmet clinical needs, the requirement for sensitive and specific biomarkers for early detection, surveillance, prognosis and drug monitoring, the role of the microbiome and the goals of personalized/precision medicine, discussing how emerging omics technologies can further inform on these areas. Exemplars from recent onco-proteogenomic-related publications will be given. Finally, we will address future perspectives, not only from the standpoint of perceived advances in treatment, but also from the hurdles that have to be overcome.

## 1. Introduction

Cancer was responsible for almost 10 million deaths in 2020 [1], making it the second most common cause of death worldwide. Currently 9/10 of the top pharma companies have a focus on cancer therapeutics, breast cancer (BC) being the leading target. Whilst there has been success in reducing mortality in some cancers [2], the updated Globocan 2020 report released by the International Agency for Research on Cancer (IARC) indicates that the global cancer burden has risen to 19.3 million cases and is predicted to rise to 30.3 million cases by 2040. Cancer is a complex, heterogeneous disease modulated by a wide spectrum of factors, such as genetic, molecular, cellular, tissue, population, environmental and socioeconomic factors, which evolve with time. Faced with this multifaceted global health issue, many research efforts have focused on the underlying disease biology and the development of innovative new treatments. Whilst the traditional “one-size-fits-all” non-precision approach to patient care using surgery, chemotherapy, radiotherapy and immunotherapy has achieved some therapeutic efficacy, many problems still have to be overcome, including recurrence [3], often associated with drug resistance [4], which facilitates tumor metastasis [5,6] and eventually promotes cancer progression [7].

There is now an emerging paradigm shift in cancer treatment, namely personalized/precision medicine. In this approach the therapeutic regimen is optimized based on a comprehensive understanding of the patient’s individual systems biology [8] with respect to both health and disease (Figure 1). This includes the compilation of comprehensive data covering their complete medical history including genetic, phenotypic, lifestyle and psychosocial characteristics to determine the most suitable therapeutic schedule and possible prognosis. This has been largely facilitated by the detailed analysis of the human genome and proteome [9,10,11], and as predicted, corresponding technological advances [12]. Proteomics is defined as the characteristic analysis of the whole protein complement of a cell, tissue or organism at a particular condition, including protein interactions, posttranslational modifications and localization [13]. Proteins are the molecules directly responsible for life, driving the structure, function and regulation of the body’s tissues and organs. While the genome is relatively static (currently 19,773 predicted proteins [14]), the proteome is extremely dynamic [15]. This is due to splice variants, PTMs (e.g., glycosylation, phosphorylation, acetylation, methylation, ubiquitination and farnesylation), often with multiple modifications and for some proteins (e.g., immunoglobulins and T-cell receptors) somatic recombination that modulate their function or activity. Due to transcriptional and translational control, not every gene is transcribed and not every RNA is translated. Additionally, stable proteins can often outlive the transcripts from which they were derived. Thus, the proteome contains >1000-fold more cellular information than the genome, with >100,000 transcripts, and potentially millions of protein variants (proteoforms) due to alternative splicing (AS), single amino acid polymorphisms (SAPs) and extensive posttranslational modifications (PTMs) [16], documenting the alterations at the level of cell, tissue or organism over time. Proteomics therefore offers a powerful resource for studying the underlying systems biology associated with health and disease, revealing potential biomarkers and drug targets. Thus, the combination of precision medicine and proteomics empowers precision oncology with increased chances for the understanding of the complex mechanisms of carcinogenesis and therapeutic targets at the molecular level, revealing potential new biomarkers for detection and surveillance and enabling new means of evaluating therapeutic efficacy and toxicity. An interesting example is that of cancer plasmonic photothermal therapy (PPTT), where plasmonic nanoparticles take light energy and convert it into localized heat, causing apoptosis in cancer cells. Variations in surface-enhanced Raman scattering (SERS) spectra, associated with MS-based metabolomics and proteomics, helping reveal the cell death mechanisms during PPTT [17].

The Proteome Reference Library (HPRL; https://hupo.org/HPP-HPRL/ accessed on 20 May 2021) [11] developed by the Human Proteomics Project (HPP) as part of the Human Protein Organisation (HUPO), reveals 24,390 cancer-related publications since 1988 using the search terms (cancer[Title/Abstract] OR oncology[Title/Abstract]) AND (HUPO[Title/Abstract] OR “human proteome project”[Title/Abstract] OR C-HPP[Title/Abstract] OR B/D-HPP[Title/Abstract] OR “missing protein”[Title/Abstract] OR proteome[Title/Abstract] OR dark proteome[Title/Abstract] OR proteomic[Title/Abstract] OR proteogenomic[Title/Abstract] OR “mass spectrometry”[Title/Abstract] OR neXtProt[Title/Abstract] OR PeptideAtlas[Title/Abstract] OR “mass spectrometric”[Title/Abstract]). These publications address cancer and proteomics (oncoproteomics) and can inform on biomarkers, cancer biology, drug discovery and therapeutic mechanisms. In the following sections we will highlight some of the key exemplars emanating from the more recent findings.

## 2. Proteomics, the Current Status

Since the term proteomics was first coined in 1994 by Mark Williams while a doctor of philosophy student at Macquarie University in Sydney, Australia [18], the technology has seen many exciting developments. Immediately coming with the initial announcement of the Human Genome Project, it was realized that it was essential to populate the human proteome for a comprehensive cognizance to the pathophysiologic mechanism behind human health and disease, using that knowledge to advance health treatment [19], with cancer recognized as a major priority. With this goal, a number of initiatives were developed including The Human Protein Organization (HUPO), The National Cancer Institute’s Clinical Proteomic Tumor Analysis Consortium (CPTAC), The Early Detection Research Network (EDRN) and SEER cancer database, The Applied Proteogenomics Organizational Learning and Outcomes (APOLLO) network and The International Cancer Proteogenome Consortium (ICPC: Cancer Moonshot). More recently, companies such as Grail (www.grail.com: proteomics accessed on 1 March 2021), Freenome (www.freenome.com: multiomics accessed on 1 March 2021), SomaLogic (www.somalogic.com: aptamer technology accessed on 1 March 2021) and Olink (www.olink.com: Proximity Extension Assay accessed on 1 March 2021) have been established.

HUPO was created in 2001 with the goal of “Translating the code of life” for a deep understanding of biology by boosting the evolution of proteomics through enhanced international cooperation, facilitating the development of advanced technologies. In 2010, the HPP was launched ensuring quality guarantee, data sharing, global cooperation and high stringency annotation of the genome-encoded proteome. The HPP has two separate approaches: chromosome based (C-HPP) and biology and disease based (BD-HPP) backed up by four pillars: mass spectrometry resources, antibody technologies, knowledgebase (bioinformatics) and, more recently (2018), pathology. The human proteome is currently at >90% completion [11].

Mass spectrometry remains the key platform currently used for proteomics analysis, with shotgun proteomics or bottom-up the most frequently utilized mode. MS-based proteomics relies on success in three main areas: sample pretreatment and analysis and data analysis. Two-dimensional gel electrophoresis (2-DE) and sodium dodecyl sulfate polyacrylamide gel electrophoresis (SDS-PAGE) were the original mainstays for sample separation before MS analysis, with the ability to separate over 10,000 proteoforms [13], and indeed these systems are still in use [20]. In this example, proteome information of tumor tissues and normal tissues was obtained by SDS-PAGE for a comparative proteomic analysis of different stages of BC. A gel-eluted liquid fractionation entrapment electrophoresis (GELFREE) system was used to separate and fractionate extracted proteins.

More recently chromatographic methods have been well recognized as methodologies worthy of consideration with particular advantages, especially in the areas of sample manipulation, recovery and automation. Multidimensional purification has been found to be particularly efficacious, giving high purification factors and reducing sample complexity prior to MS analysis, enabling deeper mining of the proteome [13,21,22,23,24]. As exemplars, Kaur et al. have designed a simple fractionation workflow to extend the coverage of the plasma proteome [25]. In a similar approach, Ahn et al. [26] used a combination of high abundance protein ultradepletion (Agilent MARS-14) and an in-house IgY depletion column, multidimensional peptide fractionation (SCX, SAX, high pH and SEC) and sequential window acquisition of all theoretical mass spectra (SWATH-MS) to screen and identify biomarkers that showed expression alterations in colorectal cancer (CRC) tissues to healthy controls.

There have been many instrumental advances over recent years, with improvements in mass accuracy, speed and resolution. More powerful MS instruments such as the Q-TOF, TOF/TOF and the Orbitrap have been developed allowing deep mining of the proteome in time frames from tens of minutes to a few hours [27]. In particular techniques for sensitive quantitative analysis have matured. In data dependent analysis (DDA) the sample is digested into peptides, ionized and analyzed by MS. In targeted proteomics (selective reaction monitoring (SRM), multiple reaction monitoring (MRM) and parallel reaction monitoring (PRM)), proteotypic peptides representing proteins of interest are used to develop rapid and sensitive assays for proteins, or panels of proteins, of interest [28]. This is particularly suited for biomarker analysis, and a compendium has been developed [29], which describes protocols for quantitation of over 99% of the annotated human proteins. However, the current method of choice is becoming data independent analysis (DIA) [30], in particular SWATH-MS [31]. In this approach, peptides within a defined mass to-charge (*m*/*z*) window are fragmented. As the mass spectrometer covers the full m/z range, repeated analysis is able to be realized, collecting the total proteome content. Additionally, much experimental evidence supports its excellent inter- and intra-laboratory reproducibility [32]. The characteristics of these approaches, including their strengths and limitations, are summarized in Table 1 and further examples related to oncoproteomics are given below.

### 2.1. iTRAQ and Other Labelling Strategies

iTRAQ enables the relative and absolute quantitation of proteins and peptides by labelling samples with isotope encoded reporter ions, allowing differential expression of proteins of interest between samples to be determined. Using iTRAQ, hundreds of proteins can be quantified and identified concurrently in a single experiment where samples labeled with 8-plex iTRAQ reagents is possible [42]. iTRAQ-based proteomics has been widely used in cancer proteomics for the analysis of complex samples like plasma [43]. For example, Serada et al. applied iTRAQ-based proteomics to study inflammatory autoimmune disorders using a comparative screen and found a novel serum biomarker, leucine-rich α-2 glycoprotein (LRG) by comparing sera samples from rheumatoid arthritis (RA) patients before and after anti-TNF therapy. Interestingly, serum levels of LRG were also related to Crohn’s disease (CD) [44]. LRG1 has been found to be highly expressed in CRC, acting as a tumor promoter [45]. The ability to detect indolent prostate cancers from those likely to progress is an important unmet clinical need. In a recent example, using a PTEN gene-knockout mouse model of prostate cancer and 8-plex iTRAQ analysis combined with transcriptomics profiling, Zhang et al. [46] found remarkable macromolecular signatures and revealed key pathway nodes, which shed light on the pathological mechanism behind prostate cancer driven by PTEN-loss, hinting at a potential valuable study direction for prostate cancer intervention.

In a similar approach, tandem mass tag (TMT) technology enabled multiplexing capabilities for quantitative proteomics analysis by labelling isobaric chemical tags on up to 10 groups of samples. Combining LC and MS, Lin et al. [47] analyzed the protein profiles of biopsy samples from patients with thyroid papillary microcarcinoma. Using quantitative analysis, important biological pathways and functional characteristics were revealed. A novel mass-defect-based carbonyl activated tag (mdCAT) enabling DIA quantification of eight samples in parallel in a single injection has recently been reported [48]. This was applied to the analysis of serum to assay the expression difference of proteins from healthy individuals and hepatocellular carcinoma (HCC) patients. An integrated proteomics workflow, combining iTRAQ, TMT and targeted approaches (MRM and PRM) for the identification and validation of potential biomarkers was reported by Kumar et al. [49].

Metabolic labeling is another alternative to chemical approaches for in vitro studies. Adding amino acids with isotope labels into cell culture, it is possible to detect proteome alterations in different states (e.g., changes in protein level during cell differentiation, protein turnover, dynamic changes of protein PTMs and interactomics) [50]. Stable isotope labeling by/with amino acids in cell culture (SILAC) detects differences in protein abundance between samples using non-radioactive isotopic labeling. To improve quantitation, Super SILAC was developed in which a mixture of SILAC-labeled cells is added as a spike-in standard for accurate quantification of unlabeled samples, thereby enabling quantification of human tissue samples, increasing its application to clinical diagnostics [51]. Cuomo et al. revealed novel biomarkers associated with tumor classification in BC by applying Super SILAC to enable multiple analysis of histone posttranslational modifications [52].

### 2.2. Targeted Approaches

Targeted proteomics approaches facilitate the development of high throughput sensitive, reproducible and quantitative assays for the measurement and validation of potential biomarkers and biomarker panels. Targeted approaches have been reported to be 5 - 10 fold more sensitive than DDA [53] and sensitivity can be further increased using immuno-enrichment [54,55]. It is worth noting that for immuno-MS the specificity requirements for the antibodies used are less stringent than for ELISA as the ultimate specificity is obtained from the fragmentation patterns of the proteotypic peptides used. The clinical potential of targeted proteomics has recently been reviewed [56,57]. It is perhaps not surprising that such assays have been used extensively. The following examples illustrate the use of different biological samples. An MRM assay, measuring the expression levels of HER2 in about 200 tissue samples, has been developed to differentiate HER2 status in BC, which is related to a worse prognosis. This assay performed better than current immunohistochemistry (IHC) methods [58]. MRM has also been used for multiplex analysis of CRC-associated proteins in human feces [59]. Using fecal samples from CRC patients and healthy volunteers, the small-scale MRM assay showed great potential for multiplex analysis in CRC. The assay was sufficiently sensitive to measure CEACAM 5, which is well known to be related to CRC, at the ng/mg feces level. The use of fecal samples for gut-related pathologies offers several advantages over other clinical biospecimens (e.g., plasma or serum) as a source of CRC biomarkers as collection is noninvasive, the test can be performed at home, one is not sample limited, and the stool effectively samples the entire length of the inner bowel wall (including any tumors or polyps present) as it passes down the gastrointestinal tract. MRM has also been used to validate seven potential urinary protein biomarkers for HCC [60].

### 2.3. Label Free Approaches

There are several benefits to label free approaches. In label-free experiments any sample can be directly compared with any other, whereas in labelled experiments it is typically only possible to directly compare samples that were physically mixed and measured in one run. Additionally, there is evidence that label-free methods achieve high coverage of the proteome as they have a higher dynamic range of quantification, allowing the exploration of low abundance proteins [61]. Some examples of the success of label free approaches are given below.

Tan et al. from the University of Hong Kong, revealed a novel mechanism of immune escape in HCC cells using label free proteomics. This showed that the immunosuppressive function of lysyl oxidase-like 4 (LOXL4) on macrophages relied primarily on PD-L1 activation. Elevated levels of LOXL4 were found to correlate with poor survival of HCC patients. Thus, in this study, proteomics shed light on the molecular mechanism of LOXL4 during the development of HCC, and also provided new ideas for possible therapeutic intervention [62].

A number of SWATH-MS papers warrant mention. Guo et al. [63] presented a SWATH-MS method for acquiring detailed proteome data from small clinical specimens such as tissue biopsies using a combination of pressure cycling technology (PCT) for efficient sample extraction [64] followed by SWATH-MS. Importantly the resulting spectral maps can be archived and reanalyzed ad infinitum using alternative search functions. In another example, Hallal et al. [65] addressed improving outcomes for diffuse glioma patients, another unmet clinical need, through the proteomics analysis of extracellular vesicles (EV) which showed that that EVs are nanoparticles with the ability to carry oncogenic molecules into the circulation against the blood–brain-barrier. SWATH was used to analyze plasma EVs isolated from preoperative glioma grade II–IV patients or controls. An 8662-protein custom library was used for data extraction. Importantly, plasma-EV protein profiles were found to cluster in line with glioma histological-subtype and grade. Analysis of EVs from patient’s plasma with recurrent tumor progression was related to more aggressive glioma samples.

The prevalence of pancreatic ductal adenocarcinoma (PDAC) is increasing globally and PDAC has the lowest survival rate of all major cancers [66]. An unmet clinical need is the ability to identify patients who do not benefit from highly morbid surgical resection, which is currently the only curative intent option. These patients could then be offered palliative chemotherapy instead. Sanhi et al. [66] used SWATH-MS to identify a plasma biomarker associated with PDAC prognosis.

Currently there are no targeted therapeutic modalities for triple negative breast cancer (TNBC), which is associated with a poor prognosis and clinical outcome. Identification of novel specific TNBC biomarkers for screening and therapeutic purposes is therefore an urgent clinical need. A recent publication [67] used silver, gold and magnetic nanoparticles to form a protein corona [68] from patient sera. The retained proteins were then separated by SDS-PAGE and analyzed by LC–MS/MS. Potential biomarkers were validated by SWATH analysis using total serum samples from TNBC patients and disease-free controls. For further examples of DIA/SWATH, and an assessment tool for the quality control of spectral libraries, readers are directed to the following excellent articles [31,69,70].

In summary, the emerging proteomics toolbox provides an excellent framework to probe cancer-related proteomes, providing the unrivalled potential to quantitatively analyze interacting proteins and their modifications, providing a blue print for understanding cancer biology. Such studies are expected to empower initiatives such as the Cancer Moonshot [71,72] and a protein equivalent to the cancer dependency map [73].

### 2.4. Proteogenomics

Proteogenomics probes the interface between proteomics and genomics [74]. The information-flow from genome to proteome involves merging the significant combination from proteomics with other omics platforms (e.g., genomics, epigenomics, transcriptomics, proteomics, lipidomics, glycomics, metabolomics and microbiomics) (Figure 2). This can provide comprehensive information on health and disease, advancing our understanding of pathophysiology, providing potential biomarkers for disease detection and surveillance, and facilitating basic and clinical cancer research for precision oncology [72].

CPTAC has invested substantial resources in proteogenomics, greatly accelerating the understanding of the molecular basis of cancer and accelerating the pace of proteogenomic research and precision oncology, with a number of publications addressing a range of cancers [11,75,76]. For example, a recent proteomic analysis of 122 treatment-naive primary breast cancers carried out by researchers from organizations including Baylor College of Medicine, Massachusetts Institute of Technology, Harvard University and CPTAC has provided one of the largest studies to date profiling the biological complexity of breast cancer. TMT-based proteomics including acetylproteome and phosphoproteome profiles combined with next-generation DNA and RNA sequencing was used to analyze primary breast cancers samples, shedding light on cell cycle progression, immunogenicity of tumors, abnormal metabolism and heterogeneity of therapeutic targets [77]. These data challenged conventional breast cancer diagnosis and provided new insights into precision/personalized medicine. In another study, using a similar workflow, proteogenomic characterization revealed therapeutic vulnerabilities during the treatment of lung adenocarcinoma and allowed the identification of differentially expressed proteins with potential diagnostic and therapeutic utility [75].

CPTAC has also made significant contributions to the establishment of CPTAC Data Portal, a Proteogenomic Cancer Atlas, which serves as the NCI’s largest public repository of proteogenomic comprehensive sequence datasets [78]. Another noteworthy database is LinkedOmics [79], which is freely available. By integrating MS-based global proteomics data generated by CPTAC on selected TCGA tumor samples (32 cancer types and a total of 11,158 patients), LinkedOmics is a very practical database for human cancer studies. By "sharing and reusing", these databases should accelerate scientific discovery and its clinical translation to patient care [79]. In what can only be described as a technological “tour de force”, Xu et al. [80] performed a comprehensive multiomics analysis (proteomics, phosphoproteomics, transcriptomics and whole-exome sequencing analysis) on 103 patients with lung adenocarcinoma (LUAD). Integrative data analysis revealed a number of cancer-associated characteristics, including protooncogene EGFR mutations, differences of proteins PTM, tumor-associated protein variants and clinical outcomes. Proteome-based classification of LUAD uncovered three subtypes (S-I~III) with distinct molecular features and a clinical phenotype.

With the ability to capture both transcript and protein information, proteogenomic profiling of healthy and tumor-derived organoids, which captures the in vivo characteristics of the original tissue in a three-dimensional in vitro culture system, can inform on the mechanisms underlying the physiopathology of tumorigenesis leading to the development of novel translational medicine strategies for cancer treatment. As an exemplar, a recent study has presented a proteogenomics analysis of human colorectal tumors and healthy organoids derived from seven patients [81]. The results show distinct signatures between organoids from different patients with patient-specific features that correlate with clinical diagnosis facilitating the development of personalized therapies [82]. A perceived limitation in the proteomics analysis of organoids has been the use of Matrigel as a scaffold material, which causes severe ion suppression due to contaminants present in the preparation. However, this was overcome in a recent study by introducing a precipitation step [83].

### 2.5. Bioinformatics

Bioinformatics plays a central role in the downstream analysis of the large body of proteomics data that is currently being generated, and as such it is one of the 4 HUPO resource pillars [11]. A number of iterative bioinformatic tools and web servers have been developed to assist in this analysis [84,85], some targeted specifically for cancer (e.g., Perseus [86] and the Cancer Genome Atlas (TCGA) [87]).

As exemplars, Dunn et al. used Perseus, the ingenuity pathway analysis (IPA^®^) and the Database for Annotation, Visualization and Integrated Discovery (DAVID) to annotate the expression and function of proteins and identified potential biomarkers and therapeutic targets of meningiomas [88]. Da et al. used bioinformatics-assisted proteomics to screen and identify the potential prognostic biomarker calcium/calmodulin-dependent serine protein kinase (CASK) in primary cholangiocarcinoma (CCA) tissues and paired precancerous tissues from surgery. Patients with negative CASK expression were found to have worse overall survival (OS) and recurrence-free survival (RFS) than those with positive CASK expression. Univariate and multivariate analyses showed that negative CASK expression was an independent risk factor for OS and RFS in CCA patients [89].

## 3. The Hallmarks of Cancer

### 3.1. Ten Hallmarks of Cancer

As recognized by Hanahan and Weinberg, understanding the intricate processes that drive normal human cells to transform into highly malignant derivatives is essential to win the "battle" against cancer [12]. They have proposed that there are a number of acquired capabilities that are shared by most, if not all, of the more than 100 types of human malignancy: the important concept of “Hallmarks of Cancer” (HoC) [90]. These hallmarks comprise: sustaining proliferative signaling, evading growth suppressors, resisting cell death, enabling replicative immortality, inducing angiogenesis, activating invasion and metastasis, reprogramming of energy metabolism and evading immune destruction [90] (Figure 3), resulting in the activation of proto-oncogenes or the silencing of tumor suppressor genes. Recent advances in proteomics and other omics technologies (speed, sensitivity, sample processing, microfabrication and automation) mean there is now an effective and efficient toolbox with which to address in depth the biology behind the HoC [91,92]. Two approaches in particular have been particularly effective in this approach: the identification of signaling pathways (interactomics) involved in disease initiation and progression, often leading to novel drug targets and the identification of potential disease related biomarkers and biomarker panels.

Methionine adenosyltransferase 2a (MAT2A) is a key metabolic enzyme in metabolism. It was reported that the deletion of MAT2A will result in proliferation inhibition in cancer cells. Thus, the team at Agios Pharmaceuticals in the US demonstrated the mechanism by which MAT2A inhibition induces DNA damage and mitotic defects in MTAP-deleted cancers using RNA sequencing and proteomics. These data shed light on the way cells evade apoptosis and sustain proliferative signaling, and provided a schema for the use of MAT2A inhibitors combined with antimitotic taxanes [93]. Activation of the telomere maintenance mechanism and instability of the nuclear genome are two key hallmarks of cancer. However, the underlying oncogenic mechanisms are still not fully understood [104]. Liu and his colleagues revealed the mechanism by which human telomerase reverse transcriptase (hTERT) upregulates and promotes cancer progression using a proteomics approach and provides a novel rational therapeutic target for HCC. In this study, a novel hTERT promoter-binding protein RBFOX3 (RNA binding protein fox-1 homolog 3) was identified using MS analysis of hTERT promoter binding protein pulldowns, which also interacted with AP-2β to regulate the expression of hTERT, exhibiting a “moonlighting function” [94]. Another proteomics study elucidated the oncogenic mechanism from the perspective of genomic integrity. Multidimensional protein identification technology (MudPIT) showed that human MMS19, the key component of the CIA targeting complex, could interact specifically with proteins related to methionine biosynthesis, DNA replication, DNA repair and telomere maintenance. These proteomics studies lay a theoretical foundation for the discovery of suitable drug targets for further pharmacological research [95].

A key control on the development of malignant tumors, which typically have poor prognosis, is the activation of invasion and metastasis, which acts in conjunction with three other hallmarks (sustained angiogenesis, evading immune response and tumor-promoting inflammation). It is now realized that the acquisition of these hallmarks is further controlled by contributions from the tumor microenvironment, consisting of the extracellular matrix [90], stromal cells and immune cells, but the precise dynamics between the nontumor microenvironment (NTME), tumor microenvironment (TME) [105] and the systemic immune system remains abstruse. A high-dimensional proteomic and transcriptomic approach was used to examine this in HCC. The data confirmed the presence of an immunosuppressive gradient in the peripheral blood, NTME and TME in primary HCC, which can regulate the activation status of tumor infiltrating leukocytes and make them immunocompromised against tumor cells [106]. The impact of the microenvironment has also been noted for brain cancer. Proteomic analysis showed cell migration-inducing and hyaluronan-binding protein (CEMIP) was increased in exosomes from metastatic brain cells and predicted metastatic progression and patient survival [96].

Previous studies have provided evidence for the role of the phosphoproteome on prognostic survival in ovarian cancer [107], but the dynamic changes in the proteome involved in tumor metastasis have not been investigated in detail. Eckert et al. used ultra-high-sensitivity mass-spectrometry-based proteomics to explore high-grade serous ovarian carcinoma (HGSC) metastasis and revealed the integrated role of the tumor stroma. Tumor and stromal compartments from surgical samples were microdissected and proteins were extracted and analyzed using an optimized proteomic workflow. Nicotinamide N-methyltransferase (NNMT) in the stroma was found to be the key molecule regulating HGSC metastasis by mediating the differentiation of cancer-associated fibroblasts (CAF). NNMT, the key metabolic enzyme regulating the differentiation of CAF and cancer progression, is a potential therapeutic target for HGSC metastasis [108]. Induction of angiogenesis is another HoC. A recent proteomics study has revealed the potential of fatty acid synthase (FASN) blockade to not only induce ovarian cancer cell death, but also exert an antiangiogenic effect [97].

The cell surface urokinase plasminogen activator receptor (uPAR) is increased in many cancers, especially in non-small cell lung cancer (NSCLC) and CRC. Levels correlate with poor prognosis and early invasion and metastasis, and currently there are several BC and prostate cancer clinical trials underway targeting either uPAR or suPAR (soluble uPAR) [98]. Integrating with the cancer hallmarks analytics tool (CHAT) analysis, a comprehensive global and plasma membrane approach (whole cell lysis with two membrane protein enrichments) using HCT116 cells (derived from Dukes’ stage D CRC) and engineered mutants with reduced uPAR expression followed by IPA analysis demonstrated that uPAR resists most pathways related with HoCs, including sustaining proliferation, evading apoptosis and metastasis [98].

Inappropriate localization of proteins is another factor responsible for oncogenesis and development by destroying normal cell function. Comparison of the nuclear proteome and transcriptome of acute myeloid leukemia (AML) blast cells with CD34+ cells from healthy human identified eleven transcription factors with abnormal expression, and remarkably affected transcription regulation. Among them, S100A4 was proposed as a therapeutic target in acute myeloid leukemia [109].

### 3.2. Emerging Hallmarks of Cancer

Redox signaling and autophagy are two emerging hallmarks of cancer. Metabolic reprogramming causes increased accumulation of reactive oxygen species (ROS), thus inducing oxidative stress, which can drive the entire process of tumorigenesis and transformation from a normal cell to a tumor [110,111]. ROS within a certain threshold range can play an important second messenger function and contribute to the fine regulation of important signaling pathways, including autophagy and apoptosis. Excessive oxidative stress can cause oxidative damage of macromolecules, leading to lipid peroxidation, DNA damage and alterations in the structure and function of proteins, which are profoundly associated with cancer initiation and progression [112,113]. Not surprisingly, some oncologists put forward the view that cancer is a redox disease, caused by an imbalance of electrons [114]. Based on the important regulation of redox in tumorigenesis, redox proteomics approaches have been well-developed to not only contribute to revealing the function and redox-modifications of important redox proteins but also provide clues for directed cancer therapy [113,115].

A sensitive and efficient proteomics technology (OxiodoTMT) has been established, which revealed a key virus-induced tumorigenesis-related redox protein, SOCS3, in HCC. These data showed that HBV-induced mitochondrial ROS production leads to episilencing of SOCS3 gene expression through snail-mediated epigenetic silencing, leading to sustained activation of the IL-6/STAT3 pathway, which ultimately contributes to hepatocarcinogenesis and indicates that SOCS3 is a potential biomarker for clinical prognosis for HCC [99]. A redox proteomics method (Oximouse) has also been developed to analyze and characterize cysteine oxidation data in mice, leading to the deepest quantitative analysis of the redox-regulated cysteine proteome to date, and generating a disease- and tissue-specific map for guiding the analysis of ROS in human diseases, especially cancer [100].

Although redox mechanisms are associated with disease, capturing much attention from researchers, the importance of another HoC, autophagy, cannot be underestimated. Autophagy exerts a dual role in tumorigenesis, which is determined by the type, stage and genetic background of the cancer [116]. To date, there have been a number of proteomics reports that shed light on the implicated relationship between autophagy and tumorigenesis, indicating potential cancer therapeutics by targeting autophagy [117]. Recently Bryant et al. proposed a promising effective dual blockade treatment for PDAC, which has a 5-year survival rate of only 9% and is characterized by autophagy-dependent growth. They concluded that blocking both the ERK/MAPK pathway and the responsive autophagic processes is effective in PDAC therapy [101]. This was supported by data generated using reverse-phase protein microarray (RPPA).

## 4. Cancer Biomarkers: Detection, Surveillance and Drug Efficacy

A biomarker may be defined as a specific characteristic that can be measured as an indicator of normal biological processes, pathogenic processes or responses to an exposure or intervention [118]. Biomarkers can be used as screening or diagnostic tools, for staging and classifying the extent of disease, defining prognosis, stratifying treatment regimens or monitoring the clinical response (e.g., drug resistance) to an intervention. Effective cancer biomarkers are therefore of great significance for cancer diagnosis and treatment, and the discovery and translational application of cancer biomarkers/biomarker panels has therefore been the main focus of many proteomics studies and the subject of numerous publications. We will illustrate this with examples relating to some of the most common and aggressive cancers.

Cancers of the pancreas have the lowest survival rate [2] and over 50% the patients present with distant metastases at diagnosis, for which the 5-year survival rate is only 9% [119]. If sensitive and specific diagnostic biomarkers for early pancreatic cancer could be found and applied, it would have a significant impact in reducing the mortality of the disease. As early as 2004, several proteins identified by 2D-PAGE and MS (e.g., S100A8, cyclophilin A, 14-3-3ζ, galectin-1, annexin A4, TM2 and peroxiredoxin I) were found to be modulated in pancreatitis tissues [120]. Later, Honda et al. compared the plasma protein profiles of pancreatic cancer patients with those of healthy volunteer through QTOF (quadrupole time-of-flight) MS, and suggested the use of apolipoprotein-AII (APOAII) isoforms (especially APOAII-2) for pancreatic cancer surveillance [121]. In a further study, they developed an ELISA for measuring the level of APOAII-2, and performed multi-institutional validation of the usefulness of APOAII-2 as a screening biomarker for pancreatic cancer [122].

CRC is one of the most common cancers in the world, and is a leading cause of cancer-related death. If detected early, it can essentially be cured by simple surgical resection (5-year survival >90%). However, by the time metastasis has occurred (20–25% of CRC cases are diagnosed at this stage) prognosis is poor with an estimated 5-year survival of only 8% [123]. Mori et al. using iTRAQ in a comparative proteomics approach, demonstrated that high expression of ezrin is related to lymph node metastasis in CRC [124]. A meta-analysis of ezrin function showed that ezrin participates in tumor metastasis and invasion, and tumorigenesis by manipulating cellular activities (e.g., adhesion, motility and proliferation) [125,126]. Recent research has confirmed the pivotal role of ezrin in regulating cell migration and invasion, and indicated this protein as a novel potential target for anticancer therapeutic approaches [127].

Lung cancer causes more deaths annually than any other cancer [2]. Using LC/MS-MS analysis, Sung et al. found that expression levels of quiescin sulfhydryl oxidase 1 (QSOX1) in lung cancer tissue were significantly higher than that in neighboring normal tissues. They also found that Lewis lung cancer cells where QSOX1 had been knocked out had reduced survival, migration and invasion capabilities under oxidative stress. In addition, QSOX1 has been proved to promote tumor metastasis in mouse models. QSOX1 could therefore be a useful lung cancer biomarker as well as a potential therapeutic target for lung cancer [128].

Obtaining effective treatment for cancer is the most important clinical endpoint. However, the clinical response to anticancer agents is frequently heterogeneous, posing a major barrier to effective cancer care. If it was possible to more accurately predict response before decisions on treatment, response rates would be improved, unnecessary ineffective treatments reduced, and global health budgets reduced. However, predicting patient response to drugs is not reliable for most cancers, owing to a lack of predictive biomarkers and an incomplete understanding of the mechanisms underlying response heterogeneity [129].

In precision medicine, resistance to chemotherapy and targeted cancer therapy can be a significant problem [129,130]. The concept of multitarget therapeutics or network therapeutics was proposed to mitigate the risk of drug resistance, and could also prove valuable in prospective drug repositioning [131]. Coscia et al. found CT45 was a promising chemosensitivity mediator and immunotherapy target using phospho- and interaction proteomics, revealing the long-term survival mechanism in HGSOC. This study showed CT45-derived HLA class I peptides had great potential for tumor treatment by activating the patient’s immune system [132].

Zhao et al. compiled a large-scale compendium (Cancer Perturbed Proteomics Atlas) of perturbed protein expression profiles by profiling proteomics alterations in response to clinically antineoplastic drugs using reverse-phase protein arrays (RPPAs). This has made up for many of the major shortcomings in the interpretation of drug antitumor mechanisms and provides a resource to investigate the dependencies of treatment responses [133].

In summary, cancer biomarkers are important tools for detecting, diagnosing, treating and monitoring tumors and judging tumor prognosis (Table 2). The rapid development of proteomics research has injected new vitality into tumor marker research. By mapping human cancer proteomes, proteomics has found differentially expressed proteins in many cancer types, which are expected to become effective cancer biomarkers. Discovering new and effective cancer biomarkers through proteomics will surely promote the further development of precision medicine and provide sick patients, high-risk groups and clinicians with more precise prevention, diagnosis and personalized treatment options.

## 5. The Microbiome and Cancer

The human microbiome is an extremely large and intricate complex set of microorganisms, composed of archaea, bacteria, viruses and eukaryotes [141]. There are approximately 100 trillion microorganisms residing in humans that are present in various locations in the body, including the gastrointestinal tract, skin, nose, mouth and, in females, the vagina. They have a biomass of up to 2 kg. Recent advances in proteomics have the potential for elaborately deciphering the function and structure of microbiome proteins and shedding light on the symbiotic relationship between humans and these microorganisms, which coevolve under normal circumstances to form a superorganism [141]. However, unexpected and uncontrolled circumstances can disrupt the body’s homeostasis, inducing the initiation and development of many diseases, including cancer (Figure 4) [142]. Although cancer is a multifactorial disease, accumulating data illuminate the powerful role of microbiota in promoting tumor growth. Recently scientists at The Weizmann Institute in Israel analyzed the microbiome from 1526 tumors and paired adjacent healthy samples from seven cancer types, namely brain, breast, ovary, melanoma, lung, pancreas and bone, shedding light on cancer progression from a microbiome perspective [143]. Increasing evidence is showing that microbes, especially intestinal microorganisms, can be both directly and indirectly implicated in cancer progression (Table 3). Exchange of nutrients with intestinal epithelial cells of the host modulates signal transduction with an impact on host metabolism, immune status and health/disease balance. Recent studies have found that a Gram-negative oral anaerobe, *Fusobacterium nucleatum* (Fn), is a significant contributor to cancer metastasis in CRC, esophageal cancer, pancreatic cancer and possibly breast cancer [144,145]. Interestingly, this observation was supported by another study, which showed that Fn, which invades the colon, stimulates tumor growth and metastatic progression in CRC by promoting cytokine secretion [146]. Another critical observation was that Fn promotes tumor metastasis through exosome secretion in CRC cells. MiRNA sequencing and proteome analysis showed Fn infection may stimulate tumor cells to generate miR-1246/92b-3p/27a-3p-rich and CXCL16/RhoA/IL-8 exosomes that are delivered to uninfected cells to promote a prometastatic behavior [147]. This research provides new insights into the molecular mechanism of the interaction between carcinogenic bacteria and the host, highlighting the importance of eliminating Fn during oncotherapy, and guiding possible clinical therapeutic approaches.

Perhaps surprisingly, rapid, accurate and cost-effective clinical methods for the routine analysis of a wide range of microorganisms in the human microbiome using MALDI-TOF were among the first MS methods to receive FDA approval for clinical use [152]: several thousand instruments have now been placed in clinical laboratories worldwide. Novel MS clinical diagnostic applications are currently being developed (e.g., antibiotic resistance (ART) and antibiotic susceptibility testing (AST) [153]).

**Table 3 cancers-13-02512-t003:** The microbiome: cancer friend or foe.

Function	Microorganism	Type of Cancer	Mechanism	References
Cancer Therapy	*Mycobacterium bovis* BCG	Bladder	Stimulating the immune system and increasing the proinflammatory cytokines activation of cancer cells phagocytosis	[154,155,156,157,158]
	*Streptococcus pyogenes* OK-432	Lymphangioma Intraoral Ranula	Immune activation by increasing cytokine levels	[159,160,161,162]
	*Clostridium novyi*	Leiomyoma	Targeting and destroying tumor cells	[154,163,164,165]
	*Salmonella Typhimurium VNP20009*	Melanoma, Pancreatic	Helping antitumor drugs target cancer	[166,167]
	*Magnetococcus marinus*	-	Targeted transport vector	[168,169]
	*Bifidobacterium Longum*	Colorectal	Enhancing the body’s immune function and regulating the expression of tumor-related genes and cytokines	[170,171,172,173,174]
	*Listeria Monocytogenes* LADD strain	Cervical, Oropharyngeal, Pancreatic, Lung and Mesothelioma	Targeted transport vector	[175,176,177]
	*Escherichia Coli*	-	Targeted transport vector	[178,179]
Tumor Promoters	Gram-Negative Bacteria	Liver Colorectal	TLR2 and TLR4 mediated upregulation of Innate inflammation; Induction of IL-17/23 pathway cytokines	[180,181,182]
	*Helicobacter Pylori*	Gastric	Inducing inflammation	[149,183,184]
	*Clostridium species*	HCC	Production of deoxycholic acid from bile and inducing inflammation	[185]
	Enterotoxigenic Bacteroides Fragilis	Colon	Inducing inflammation	[186]
	*Fusobacterium*	Colorectal	Inducing inflammation and protecting tumors from an immune cell attack	[187,188,189]
	*Escherichia Coli*	Colorectal	PKS inducing DNA breaks	[190]
	*Chlamydia Pneumoniae*	Lung	C. pneumoniae protein interfering with host cell behavior	[191]
	*Chlamydia Trachomatis*	Cervical, Ovarian	Promotes host cell DNA double-strand breaks, induces host cell genome instability and even transformation	[192]

Several microbial proteomics studies have shown that microorganisms promote tumor growth and metastatic progression and there is increasing data indicating that the microbiome is partly responsible for the initiation of intestinal tumors. Verberkmoes et al. [193] used a non-targeted, shotgun mass spectrometry-based metaproteomics approach for the first deep proteome measurements of the human distal gut microbiota and proteomics studies on patients with CRC also identified a number of microbial proteins in their fecal samples [59]. Bosch et al. have identified new potential biomarkers for CRC screening through MS analysis of stool samples. A four protein biomarker panel had sensitivities of 80% and 45% for detecting CRC and advanced adenomas, respectively, at 95% specificity [194]. A quantitative metaproteomic study characterized the abundance differences of microbial proteins between fecal samples from CRC patients and healthy controls, illustrating the pathogenesis of CRC and showed the promising potential of metaproteomics in clinical diagnostics in the future [195]. The lung is also colonized by hundreds of bacterial species, which can impact on the progression of lung cancer. A recent proteogenomic approach uncovered the mechanism by which local microbiota provoke inflammation and promote cell proliferation in lung cancer. Superfluous commensal bacteria stimulate γδT cells to duplicate and secrete inflammatory cytokines like IL-17 and IL-2 (Figure 4), which will provide a suitable environment for tumor proliferation and survival [148]. It has been suggested the genera *Veillonella* and *Megasphaera* may act as lung cancer biomarkers with good sensitivity and specificity [196]. Increased understanding of the carcinogenic mechanism of microorganisms has encouraged researchers to consider how to utilize them for cancer treatment. Emerging data has shown antitumor activity of microbial proteins and metabolites derived from microbial activity such as microbial-derived short-chain fatty acids (SCFAs) [197] and bacterial lipopolysaccharide (LPS) [198]. Interestingly, it has been found that microbiota can act as tumor-suppressor agents. For example, probiotic-derived ferrichrome secreted from *Lactobacillus casei* was identified as an effective tumor suppressor in colon cancer by MS [150]. This study suggests that ferrichrome can activate the JNK signaling pathway triggering apoptosis (Figure 4), and is more effective with reduced adverse effects than the clinical drugs cisplatin and 5-fluorouracil. Further studies have found that ferrichrome can also restrain proliferation and induce apoptosis in other cancers, such as gastric cancer [199], HCC [200] and pancreatic cancer [201]. Fecal microbiota transplantation (FMT) is the reconstruction of the gut microbiota by transplanting fecal material from healthy donors to sick patients. It has been used as an effective mode of treatment for intestinal and extraintestinal diseases [202]. Thus, FMT has been demonstrated to be an effective therapy for human diseases including inflammatory bowel disease (IBD) [203], *Clostridium difficile* infection (CDI) [204] and major depressive disorder (MDD) [205].The first-in-human FMT cancer clinical trials were reported by Baruch et al. [206] and Davar et al. [207], who demonstrated that FMT has great potential in cancer immunotherapy in combination with anti-PD-1. However, although FMT has been approved for clinical application by the FDA, there are still many hurdles to be overcome for its safe and effective routine use. Indeed, the FDA has recently acknowledged that more careful donor testing must be undertaken following the death of a patient due to the presence of a rare drug-resistant form of *E. coli* in a fecal transplant. Proteomics can clearly play an important role in supporting FMT. It can help in the analysis of donor transplant material while metaproteomics can help identify optimal donors by comparative recipients/donor studies. It can also inform on the method of administration by analysis of the structure, function and composition of microbiota in fecal samples and can monitor the efficacy of the procedure. Moreover, proteomics coupled with genomic methods, such as DNA sequencing, can be used to survey changes in microbiota diversity, function and structure before and after FMT in order to evaluate the therapeutic effect.

The microbial ‘Dark Matter’ [208] represents the 30–40% of the microbes in the GI tract that currently cannot be cultivated or identified. This niche could contain interesting enzymes, new antimicrobials and other potential therapeutics. Microbial culturomics [209], in which MS is used in conjunction with extensive sequencing to explore the suitability of a wide range of known and novel culture methods, can help unravel this.

In conclusion, the human microbiome is a promising emerging target for both cancer development and therapeutics. It may be directly oncogenic, through promotion of mucosal inflammation or systemic dysregulation, or may affect anticancer immunity/therapy. Analysis of the microbiome may play a role in both the diagnosis and treatment of tumors [210]. As a revolutionary tool in biochemical research, proteomics can perform detailed protein profiling of the microbiota, thereby discovering potential biomarkers and uncovering altered disease related protein levels and biological pathways. Fecal proteomics, a non-invasive test, has many significant advantages for the discovery and validation of biomarkers for screening CRC.

## 6. Future Directions/Perspectives

Proteomics has made remarkable progress over the last decade, and that trend is set to continue with further advances expected in terms of speed, sensitivity, reproducibility, automation and throughput [11,211]. Developing clinically useful diagnostic and prognostic biomarkers to identify individuals needing treatment and developing predictive biomarkers that identify and select individuals who will benefit most from these therapies to support personalized/precision medicine will be a priority. Large, multidisciplinary, multinational teams will be commonplace, and rare diseases will receive more attention and increased funding. Industrial scale laboratories for large scale cancer proteome initiatives (e.g., ProCan [212], Stoller Biomarker Discovery Centre (http://www.biomarkers.manchester.ac.uk/about/sbdc/ accessed on 12 March 2021), The Chinese Pilot Hub Of Encyclopedic Proteomix (PHOENIX)) will continue to be developed (PHOENIX, using the Tianhe 2 Super Computer, can generate up to 40 proteomes in a day). Large scale health projects (e.g., Cancer Moonshot, The Stanford/HUPO http://med.stanford.edu/hpop.html hPOP Study accessed on 12 March 2021) will be undertaken. However, it is anticipated that there will be a shift away from MS-based approaches for clinical applications (e.g., microarrays, aptamer technologies and proximity assays). This will be assisted by improved antibody validation to ensure specificity [213], and the increased use of resources such as the Human Protein Atlas [214] and Antibodypedia [215]. Some of these emerging technologies are discussed below.

### 6.1. Top-Down MS

To date most of the publications on the characterization and identification of proteins have used an enzymatic or chemical digestion approach in which the proteins are fragmented into small (bottom up) or medium size (middle up) peptide fragments. However, top-down methods, in which the full-length protein is analyzed are now coming of age. Importantly this technology has the potential to analyze proteoforms that arise from alternative splicing events and/or PTMs for basic and clinical research without loss of information often observed with the digestion of proteins [216]. However, it is currently approximately 100-fold less sensitive than bottom up MS and has been reported to have reduced proteomic coverage and throughput [217].

### 6.2. Differential Ion Mobility MS (DMS)

Proteomic analyses can benefit from additional techniques to improve protein or peptide separation and achieve increased protein identification allowing deeper mining of the proteome. DMS separates ions in the gas phase based upon subtle differences in their chemical structures (e.g., mass, shape, center of mass and dipole moment), giving separative power that is orthogonal to both MS and HPLC [218]. A current limitation is that the method is relatively time consuming and data intensive [219].

### 6.3. Imaging Mass Cytometry

Multiplexed imaging methods are becoming increasingly important [220] and have been applied to oncology. As an example of this, the Bodenmillar laboratory [221] used CyTOF coupled with genomics approach to define the phenogenomic profile of BC. CyTOF combines flow cytometry with elemental mass spectrometry. In this technology, antibodies coupled to purified isotopes from rare-earth metals are used to label cells. Cell samples are exposed to an inductively coupled argon plasma (ICP) torch for vaporization, atomization and ionization. The atomic ion cloud can then be introduced into a hybrid quadrupole-TOF MS for quantitation. The current CyTOF instrumentation allows simultaneous measurement of over 40 cellular parameters with more than 100 detection channels at single-cell resolution. It is being used extensively in single cell proteomics applications [222].

### 6.4. Microarrays

Protein microarrays are being seen as versatile tools for protein–macromolecule interaction analysis and drug target identification in a rapid, reproducible and cost-effective manner. This has been facilitated by improved surface chemistry and advances in miniaturization and microfluidics [223] and is being used to support clinical applications [224]. Currently three types of microarrays are in routine use: analytical microarrays (library of antibodies, aptamers or affibodies on the surface), functional microarrays (arrays containing full-length functional proteins or protein domains) and reverse phase microarrays (RPMA: cell or tissue lysates). As exemplars, as part of an integrated genomic, transcriptomic and proteomic profiling of 150 PDAC specimens, RPMAs were used to help characterize pancreatic ductal adenocarcinoma [225]. Researchers from the MD Anderson Cancer Center have used RPMSs to characterize human cancer cell lines and analyze functional cancer proteomic data using the Cancer Proteome Atlas. A total of 8000 patient samples involving 32 types of cancer and more than 650 cell lines were addressed [226]. RPMAs have also been used to generate and compile perturbed expression profiles in the large-scale characterization of drug responses of clinically relevant proteins in the cancer cell lines [133].

### 6.5. Big Data, Artificial Intelligence (AI), Machine and Deep Learning

Data analysis is rapidly becoming the rate limiting step in proteogenomics, as large-scale proteomics projects are capable of generating terabytes of high volume, velocity, variety and veracity data on a daily basis, the analysis of which is becoming overwhelming (the big data problem) [227]. Such data are typically highly dimensional and nonlinear, making them difficult to analyze using conventional statistical methods. Integration of proteomics with artificial intelligence methods like machine and deep learning clearly represents the future trend for proteomics research and personalized/precision medicine and a number of platforms are being developed. Recent examples include applications for HCC [228], renal cell carcinoma [229] and lung cancer [230]. Researchers from the Technical University of Munich successfully used proteomic data to train a neural network, termed Prosit, facilitating the rapid and accurate error free mass analysis of proteins [231]. Another important use of AI has been in protein-structure-prediction systems, which could significantly assist drug discovery [232]. Many of the structures generated were indistinguishable from those determined by X-ray crystallography and/or CryoEM.

### 6.6. Single-Cell Proteomics

The ability to measure proteins at the single cell level offers the potential to investigate cancer heterogeneity and signal transduction and will realize many transformative opportunities [233]. A number of techniques are currently being investigated including high-resolution imaging using genetically encoded fluorescent antibody-based strategies coupled with flow cytometry, mass cytometry or microfluidics and MS [234]. Applications include mechanisms of drug resistance [235], cancer recurrence [236] and an understanding of tumor immunity [237]. Fluorescence-based systems have been extensively used due to their sensitivity, selectivity and direct readout [238,239,240]. Bioinformatics will play a key role in data analysis [241,242].

Commercial companies are now supporting single cell proteomics. For example, Isoplexus is offering single cell secretome, metabolome and intracellular proteome packages (www.isoplexus.com/ accessed on 12 March 2021) while Erisyon has been launched to commercialize a single-molecule protein sequencer based on Edman chemistry with fluorescent tags (www.erisyon.com/ accessed on 12 March 2021).

Single cell proteomics is rapidly coming of age and will give novel insights into the HoC and new therapies with the potential to transform oncology. However, at present we are just at “the tip of the iceberg”, and a number of challenges remain [243]. Foremost amongst these is perhaps finding patterns in spatially resolved measurements and the integration of single-cell data across multiple samples, experiments and types of measurement.

## 7. Conclusions

Translational studies are leading to new and improved clinical assays, which will continue to facilitate the roll out of precision medicine, guiding oncologists to find the optimum treatment for individual patients. There will almost certainly be a need for new clinical trial designs, addressing in particular tumor and patient heterogeneity in a personalized/precision medicine approach. Increased understanding of the underlying tumor biology has led to an expanding number of potential therapeutic targets leading to the development of new drugs. The use of individually tailored combinations of precision targeted drugs identified by proteogenomics will require sophisticated optimization based on new strategies (e.g., optimum delivery regimes) to ensure maximum efficacy (reviewed in [244,245]).

Novel sensitive and specific biomarkers and biomarker panels will be discovered, and new drug targets and drugs identified. It is perhaps the uptake of precision medicine that now poses the major hurdles, in particular the big data problem mentioned above, and concerns in some areas of the community about privacy, ethical responsibilities and equity, with a growing gap in health system parity some between high and low-income countries, and in some cases even different ethnic groups [246]. To address this, the role of informed precision medicine alliances and coalitions bringing together multidisciplinary international groups of thought leaders, researchers and oncologists from academia and industry (e.g., The Oncology Think Tank (TOTT), WR Worldwide Innovative Networking In Personalized Cancer Medicine and The European Personalized Medicine Association) is critical, facilitating true globalization. In this respect, the current COVID-19 pandemic has shown how rapidly medical breakthroughs can be achieved, and traditional barriers (e.g., data sharing between big pharma) broken down.

## Figures and Tables

**Figure 1 cancers-13-02512-f001:**
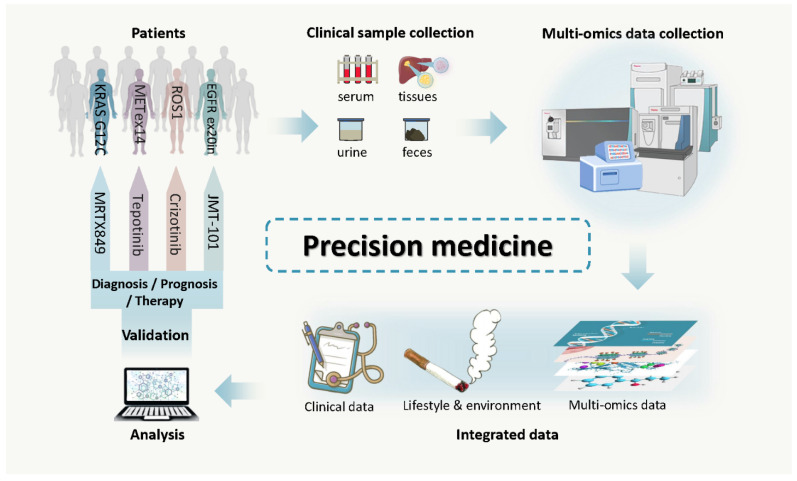
Precision medicine. Samples (e.g., serum/tissue/urine/feces) can be analyzed by multiomics technologies. Blood informs on the systemic response to the disease, urine includes the host metabolites which are excreted from the body, stool samples show what the intestine is exposed to and resected tissue can give information on the response mounted at the site of disease. Information from multiomics analyses, clinical reports, lifestyle and psychosocial characteristics can identify biomarkers for prevention, diagnosis, prognosis and surveillance, and help to determine the most suitable therapeutic schedule for individuals presenting with a given disease.

**Figure 2 cancers-13-02512-f002:**
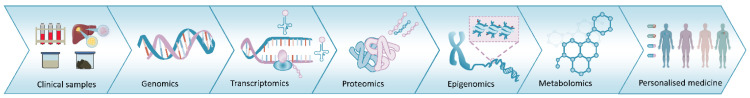
The omics pipeline. The information-flow from multiomics platforms can provide comprehensive information on health and disease, facilitating the realization of the goal of precision medicine.

**Figure 3 cancers-13-02512-f003:**
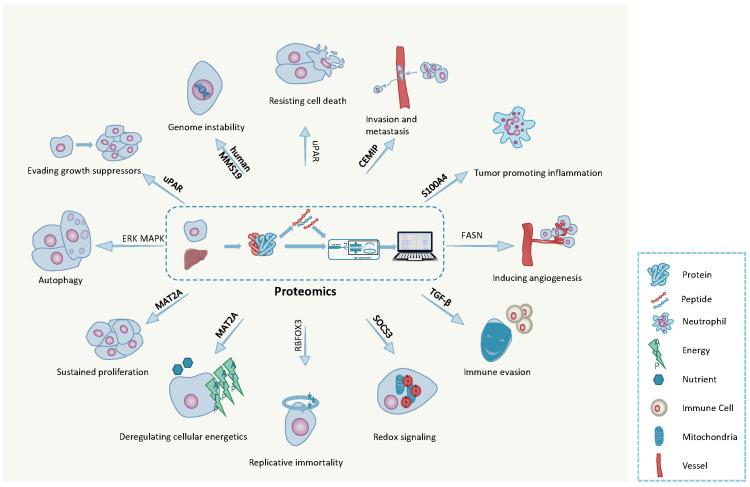
The Hallmarks of Cancer. Proteomics helps advance the understanding the Hallmarks of Cancer and makes a significant contribution to revealing the mechanisms behind cancer development [93,94,95,96,97,98,99,100,101,102,103].

**Figure 4 cancers-13-02512-f004:**
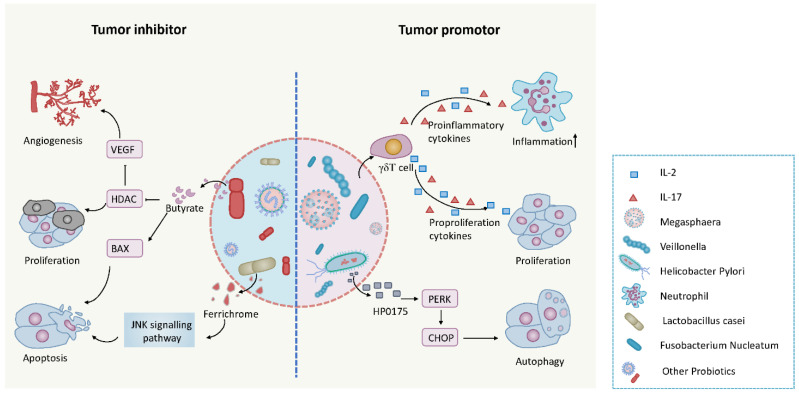
The microbiome plays a dual role in human health and disease. Microbiota and their metabolites can act as either tumor promotors or tumor inhibitors. As tumor promotors they can secrete antigens or cytokines to induce adaptive autophagy, inflammation and tumor proliferation promoting tumor progression. Microbes, especially some Gram-negative bacteria, can be directly or indirectly involved in cancer progression and regulation. For example, local microbiota can provoke inflammation associated with lung adenocarcinoma by activating lung-resident γδ T cells and promoting tumor cell proliferation by secreting IL-22 and Areg [148]. The secreted antigen, HP0175, of *Helicobacter pylori* links the unfolded protein response (UPR) to autophagy in gastric epithelial cells. *Helicobacter pylori* secrete HP0175, inducing adaptive autophagy to promote tumor cell survival [149]. As examples of tumor inhibitors, ferrichrome, secreted from *Lactobacillus casei*, can induce colon cancer apoptosis [150]. Some probiotic proteins and metabolites have antineoplastic activity. For example, butyrate shows great anti-cancer potential by inhibiting angiogenesis and proliferation and inducing apoptosis [151].

**Table 1 cancers-13-02512-t001:** Proteomics toolbox.

Proteomics Toolbox			Technical Process	Strengths	Limitations	Reference
Targeted Approaches	MRM, SRM		(1) Primary mass spectrometry scans to screen out parent ions that are consistent with the specificity of the target molecule. (2) Collision and fragmentation of parent ions to remove interfering ions. (3) Mass spectrometry signals collected from selected specific ions	(1) High sensitivity (2) Good accuracy (3) Good reproducibility (4) High-throughput	(1) Only preselected proteins can be detected (2) No screening analysis	[33,34]
	PRM		(1) Select targeted peptides. (2) Define acquisition method. (3) Preliminary experiment to adjust the parameters and target peptide fragment. (4) Perform PRM analysis. (5) Analyze data	(1) Simpler and cheaper. (2) Wider linear detection range. (3) Higher selectivity, better sensitivity and better reproducibility	(1) Only preselected proteins can be detected (2) No screening analysis (3) When the number of peptides to be analyzed is large it is necessary to fine-tune the MS collection parameters	[35,36]
Untargeted Approaches	DDA	iTRAQ/TMT	(1) Enzymatic or chemical fragmentation. (2) Differential labeling using iTRAQ/TMT reagent. (3) Mixed labeled protein samples analyzed by tandem mass spectrometry. (4) Data analysis	(1) Good repeatability. (2) Mature method. (3) Wide range of sample sources	(1) Easily contaminated by other proteins in the sample (2) Maximum 12 channels at a time	[37]
		Label-free	(1) Protein extraction. (2) Protein digestion. (3) LC–MS/MS analysis. (4) Data analysis	(1) Low cost (2) Not limited by the number of samples. (3) Wide range of application	(1) Highly dependent on machine stability. (2) Quantitative results can be unreliable	[38]
	DIA	SWATH	(1) Sample digestion (2) Consecutive, adjacent precursor ion windows (SWATHs) scanned using a Triple TOF MS (3) Data analyzed using bioinformatics and a relevant reference spectral library	(1) Good reproducibility. (2) Less affected by high-abundance proteins	Quality of spectral library for data analysis	[32,39,40,41]

**Table 2 cancers-13-02512-t002:** Examples of cancer biomarkers: discovered by proteomics and their applications in precision medicine.

Cancer Biomarkers	Cancer Type	Use	Proteomic Technology	Reference
APOAII-2	Pancreatic cancer	Detection of early-stage pancreatic cancer and risk factors for pancreatic malignancy	2D-PAGE, QTOF MS system and ELISA	[121,122]
Ezrin	Colorectal cancer	Predicting lymph node metastasis in colorectal cancer	iTRAQ in a comparative proteomics approach	[124,126,127]
QSOX1	Lung cancer	A potential therapeutic target	LC/MS-MS analysis	[128]
Serine/threonine kinase 4	Colorectal cancer	An early detection biomarker	MALDI-TOF-MS	[134]
αB-crystallin	Breast cancer	A potential prognostic biomarker	Quantitative iTRAQ proteomics	[135]
ENO1	Lung cancer	A Potential Sputum Biomarker for Early-Stage Lung Cancer and a potential therapeutic target	Shotgun proteomics; liquid chromatography–tandem mass spectrometry technology	[136,137,138,139]
CT45	High-grade serous ovarian cancer	A Chemosensitivity Mediator and Immunotherapy Target	Liquid Chromatography–MS analysis; Phosphatase activity assay and phosphoproteomics; Immunofluorescence	[132]
FoxO3a	Breast cancer	A Positive Prognostic Marker and a Therapeutic Target in Tamoxifen-Resistant	Label-Free Semiquantitative Proteomic Analysis and ingenuity pathway analysis (IPA)	[140]
HSP 90β	Lung adenocarcinoma	A potential prognostic biomarker	Nano-LC–MS/MS analysis; ELISA; label-free quantification	[80]
SAA2, APCS, APOA4, F2 and AMBP	Colorectal cancer	Potential early diagnosis biomarkers	SWATH-MS and ELISA	[26]

## Data Availability

No new data were created or analyzed in this study. Data sharing is not applicable to this article.

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
