# Peer review of "Proteomics, Personalized Medicine and Cancer"

_cancers, 2021, doi:10.3390/cancers13112512_

Round 1

Reviewer 1 Report

In this manuscript, Su and colleagues present an elegant summary of the latest findings regarding the emergent role of Proteomics approaches in personalized medicine and cancer treatments. The manuscript is well structured, clearly written, and addresses all the fundamental issues about the main proteomics tools used for precision medicine approaches in fighting human cancer. Moreover, all the information here provided are well supported with appropriate cited and commented articles. The manuscript is already suitable for publication in Cancers. However, below are minor suggestions for authors who should better address and correct some points that would make the manuscript comprehensive of highlights on proteo(geno)mics approaches in cancer research and translational medicine.

1) One of the main concerns the proteogenomics paragraph (page 8). I suggest the authors implement these highlights with the current findings obtained thanks to the proteogenomics approach for the personalized proteome profiling of health and tumor patients-derived organoids. These models represent a challenging task to study tumorigenesis's physiopathological aspects to develop a novel translational medicine strategy in cancer research.

2) In the manuscript, the authors missing to describe in-depth the relevance of important proteogenomics (LinkedOmics) cancer datasets. It represents a fundamental tool for applying the precision medicine approach in the current oncological clinical practice.

3) Page 8, line 294, citation.11: I encourage the authors to add other citations to support the previously sentence reported.

4) Page 1, line 25, Abstract section: Please, correct the font style of “which” according to the format font style required for the publication to “Cancers” journal.

5) In the list of references, some of which need to be corrected according to the format required by guidelines for publication of “Cancers” (i.e. refs: 21, 111). Please consider verifying the correct format of all extensive list carefully.

Author Response

In this manuscript, Su and colleagues present an elegant summary of the latest findings regarding the emergent role of Proteomics approaches in personalized medicine and cancer treatments. The manuscript is well structured, clearly written, and addresses all the fundamental issues about the main proteomics tools used for precision medicine approaches in fighting human cancer. Moreover, all the information here provided are well supported with appropriate cited and commented articles. The manuscript is already suitable for publication in Cancers.

However, below are minor suggestions for authors who should better address and correct some points that would make the manuscript comprehensive of highlights on proteo(geno)mics approaches in cancer research and translational medicine.

Point 1: One of the main concerns the proteogenomics paragraph (page 8). I suggest the authors implement these highlights with the current findings obtained thanks to the proteogenomics approach for the personalized proteome profiling of health and tumor patients-derived organoids. These models represent a challenging task to study tumorigenesis's physiopathological aspects to develop a novel translational medicine strategy in cancer research.

Response 1: Thank you for your useful suggestion. We have now expanded that paragraph to include this. (Page 9, line 336-348)

Point 2: In the manuscript, the authors missing to describe in-depth the relevance of important proteogenomics (LinkedOmics) cancer datasets. It represents a fundamental tool for applying the precision medicine approach in the current oncological clinical practice.

Response 2: Again, this has now been addressed. (Page 8, line 320-327)

Point 3: Page 8, line 294, citation.11: I encourage the authors to add other citations to support the previously sentence reported.

Response 3: We have added the following references: Cell. 2020;182(1):200-25; Cell. 2020;183(7):1962-85. (Page 8, line 307)

Point 4: Page 1, line 25, Abstract section: Please, correct the font style of “which” according to the format font style required for the publication to “Cancers” journal.

Response 4: This is now correct. (Page 1, line 25)

Point 5: In the list of references, some of which need to be corrected according to the format required by guidelines for publication of “Cancers” (i.e. refs: 21, 111). Please consider verifying the correct format of all extensive list carefully.

Response 5: We have been through and checked the reference format.

Reviewer 2 Report

Review by Su et al is a timely article highlighting the role of proteomics in personalized medicine and cancer. Authors have appropriately included important contributions made by the ‘omics technologies’ in addressing important components of the hallmarks of cancer, microbiome, biomarkers and therapy. The following comments/suggestions could help improve the manuscript.

  1. 1. Lines 53-55: There is now an emerging paradigm shift in cancer treatment, namely personal-

ized/precision medicine, where therapeutic regimen is optimized ground on a comprehensive understanding of the patient’s individual systems biology [8] with respect to both health and disease (Figure 1). Need to restructure the sentence to make sense.

  1. Line 65: ‘.. or organism at a particular conditions’ should be changed to …particular condition

  1. Lines 64-65: While the genome is relatively static (currently 19773 predicted proteins [14]), the proteome is extremely dynamic [15]. Sentence not clear!

  1. Line 198: were reperted---change to were reported

  1. Line 208: Alessandro Cuomo et al: only last name needs to be given here.

  1. Line 221: 200 stissue samples---change to 200 tissue samples

  1. Line 337: "Hallmarks of Cancer" (HoC); please be consistent in use of abbreviation; some places it is indicated as HOC

  1. In the section 4. Cancer Biomarkers: Detection, surveillance and drug efficacy (lines 465-476), the abbreviation ‘PC’ is being used for what? It seems the references cited are for pancreatic cancer at three spots, but authors have used ‘PC’ earlier in the manuscript to indicate prostate cancer. Please correct.

  1. Line 592: ‘ inflammatory cytokines like IL-17 and IL-2 (Figure 4): please include the names of these cytokines in the figure itself.

  1. Line 599: Please define SCFAs

  1. Line 606: PC [193].: clarify if author is referring to prosate or pancreatic cancer.

  1. Line 621-625: Change the two sentences please, these are not clear ‘What's more, information about ….. function of microbiota. ‘Furthermore, the integration of …..diversity, and function.

  1. Line 628: change Cultureomics to Microbial cultureomics.

  1. Line 632: Rewrite the sentence ‘It may be carcinogenic, …..paling anticancer therapy.
  2. Line 645: Restructure the sentence, ‘Large, multidisciplinary,….. and funding.

  1. Line 694: Incomplete sentence, ‘Researchers from the MD Anderson…..more than 650 cell lines) [216].

  1. Heading on line 700: Change ‘Arteficial Intelligence’ to Artificial Intelligence’

  1. Line 722: Reconstruct the sentence, ‘Fluorescence-based systems are becoming extensively

used [228-230].

Author Response

Review by Su et al is a timely article highlighting the role of proteomics in personalized medicine and cancer. Authors have appropriately included important contributions made by the ‘omics technologies’ in addressing important components of the hallmarks of cancer, microbiome, biomarkers and therapy.

We thank you for your very positive comments.

The following comments/suggestions could help improve the manuscript.

Point 1: Lines 53-55: There is now an emerging paradigm shift in cancer treatment, namely personalized/precision medicine, where the therapeutic regimen is optimized based on a comprehensive understanding of the patient’s individual systems biology [8] with respect to both health and disease (Figure 1).Need to restructure the sentence to make sense.

Response 1: This has been corrected. (Page 2, line 54-57)

Point 2: Line 65: ‘.. or organism at a particular conditions’ should be changed to …particular condition

Response 2: This has been corrected. (Page 2, line 63)

Point 3: Lines 64-65: While the genome is relatively static (currently 19773 predicted proteins [14]), the proteome is extremely dynamic [15]. This is due to splice variants, PTMs (e.g. glycosylation, phosphorylation, acetylation, methylation, ubiquitination, farnesylation), often with multiple modifications and for some proteins (e.g. immunoglobulins, T-cell receptors) somatic recombination which modulate their function or activity.

Response 3: Lines 64-65 have been rewritten as cited above. (Page 2, line 66-71)

Point 4: Line 198: were reperted---change to were reported

Response 4: This typo has been corrected. (Page 6, line 211)

Point 5: Line 208: Alessandro Cuomo et al: only last name needs to be given here.

Response 5: This has been corrected. (Page 6, line 221)

Point 6: Line 221: 200 tissue samples---change to 200 tissue samples

Response 6: This has been changed. (Page 7, line 234)

Point 7: Line 337: "Hallmarks of Cancer" (HoC); please be consistent in use of abbreviation; some places it is indicated as HOC

Response 7: This has now been updated for consistency.

Point 8: In the section 4. Cancer Biomarkers: Detection, surveillance and drug efficacy (lines 465-476), the abbreviation ‘PC’ is being used for what? It seems the references cited are for pancreatic cancer at three spots, but authors have used ‘PC’ earlier in the manuscript to indicate prostate cancer. Please correct.

Response 8: This has been addressed.

Point 9: Line 592: ‘ inflammatory cytokines like IL-17 and IL-2 (Figure 4): please include the names of these cytokines in the figure itself.

Response 9: This has been added to the text box in the figure.

Point 10: Line 599: Please define SCFAs

Response 10: This has been done. (Page 17, line 647; Page 22)

Point 11: Line 606: PC [193].: clarify if author is referring to prosate or pancreatic cancer.

Response 11: This has been addressed. It refers to pancreatic cancer. (Page 17, line 655)

Point 12: Line 621-625: Change the two sentences please, these are not clear ‘What's more, information about ….. function of microbiota. ‘Furthermore, the integration of …..diversity, and function.

Response 12: This has been addressed. (Page 17, line 668-675)

Point 13: Line 628: change Cultureomics to Microbial cultureomics.

Response 13:This has been done. (Page 17, line 678)

Point 14: Line 632: Rewrite the sentence ‘It may be carcinogenic, …..paling anticancer therapy

Response 14: This has been done. (Page 18, line 687-691)

Point 15: Line 645: Restructure the sentence, ‘Large, multidisciplinary,….. and funding

Response 15: Again, this has been done. (Page 18, line 698-700)

Point 16: Line 694: Incomplete sentence, ‘Researchers from the MD Anderson…..more than 650 cell lines) [216].

Response 16: This has been corrected. (Page 19, line 750-753)

Point 17: Heading on line 700: Change ‘Arteficial Intelligence’ to Artificial Intelligence’

Response 17: This has been done. (Page 19, line 757)

Point 18: Line 722: Reconstruct the sentence, ‘Fluorescence-based systems are becoming extensively used [228-230]

Response 18:This has been updated. (Page 19, line 779-781)

Reviewer 3 Report

This is a comprehensive and wide ranging review.

For balance it needs more discussion of limitations of the technology such as sensitivity and functional validation. What are the major challenges for the future for each approach.

Discuss the need for new clinical trial designs (provide examples) to address tumor and patient heterogeneity for personalized medicine.

Author Response

Reviewer 3

This is a comprehensive and wide ranging review.

Thank you for your very positive comments.

Point 1: For balance it needs more discussion of limitations of the technology such as sensitivity and functional validation.

Response 1: We had attempted to address many of the technological limitations in Table 1. This has now been highlighted. Additionally, many of the advances have been addressed as overcoming existing problems. (page 5)

Point 2: What are the major challenges for the future for each approach.

Response 2: Aside from the technological limitations, we have now specifically addressed challenges for some of the emerging technologies, including organoid analysis, top down proteomics, Differential Ion Mobility MS and single cell proteomics. (page 18)

Point 3: Discuss the need for new clinical trial designs (provide examples) to address tumor and patient heterogeneity for personalized medicine.

Response 3: We have now covered this in the Conclusions section. Two key reviews have been cited. (page 20, line 796-803)

Additionally, a number of minor grammatical corrections which we identified during the read through have been corrected.

Round 2

Reviewer 3 Report

Minimum of revisions in response to the comments. Although the comments have been responded to, I hoped for more substantial additions and novel guiding insights for the future.